# Optimization of Technological Parameters of the Process of Forming Therapeutic Biopolymer Nanofilled Films

**DOI:** 10.3390/nano12142413

**Published:** 2022-07-14

**Authors:** Michał Bembenek, Oleg Popadyuk, Thaer Shihab, Liubomyr Ropyak, Andrzej Uhryński, Vasyl Vytvytskyi, Oleksandr Bulbuk

**Affiliations:** 1Department of Manufacturing Systems, Faculty of Mechanical Engineering and Robotics, AGH University of Science and Technology, 30-059 Krakow, Poland; 2Department of General Surgery, Ivano-Frankivsk National Medical University, 076000 Ivano-Frankivsk, Ukraine; popadyukoleg@ukr.net; 3Medical Instruments Techniques Engineering Department, Technical College of Engineering, Al-Bayan University, Baghdad 10070, Iraq; thaer.a@albayan.edu.iq; 4Department of Welding, Ivano-Frankivsk National Technical University of Oil and Gas, 076019 Ivano-Frankivsk, Ukraine; 5Department of Computerized Engineering, Ivano-Frankivsk National Technical University of Oil and Gas, 076019 Ivano-Frankivsk, Ukraine; l_ropjak@ukr.net; 6Department of Machine Design and Operation, Faculty of Mechanical Engineering and Robotics, AGH University of Science and Technology, 30-059 Krakow, Poland; uhrynski@agh.edu.pl; 7Department of Engineering and Computer Graphics, Ivano-Frankivsk National Technical University of Oil and Gas, 076019 Ivano-Frankivsk, Ukraine; vytvytskyi.v.s@gmail.com; 8Department of Prosthetic Dentistry, Ivano-Frankivsk National Medical University, 076000 Ivano-Frankivsk, Ukraine; obulbuk@ifnmu.edu.ua

**Keywords:** biopolymer films, zinc oxide nanoparticles, experiment planning, mathematical model, optimization, therapeutic properties, physical and mechanical properties, degradation of biopolymers, chrome coating, ecological safety

## Abstract

The prospects of using biopolymer nano-containing films for wound healing were substantiated. The main components of biopolymer composites are gelatin, polyvinyl alcohol, glycerin, lactic acid, distilled water, and zinc oxide (ZnO) nanoparticles (NPs). Biopolymer composites were produced according to various technological parameters using a mould with a chrome coating. The therapeutic properties of biopolymer films were evaluated by measuring the diameter of the protective effect. Physico-mechanical properties were studied: elasticity, vapour permeability, degradation time, and swelling. To study the influence of technological parameters of the formation process of therapeutic biopolymer nanofilled films on their therapeutic and physico-mechanical properties, the planning of the experiment was used. According to the results of the experiments, mathematical models of the second-order were built. The optimal values of technological parameters of the process are determined, which provide biopolymer nanofilled films with maximum healing ability (diameter of protective action) and sufficiently high physical and mechanical properties: elasticity, vapour permeability, degradation time and swelling. The research results showed that the healing properties of biopolymer films mainly depend on the content of ZnO NPs. Degradation of these biopolymer films provides dosed drug delivery to the affected area. The products of destruction are carbon dioxide, water, and a small amount of ZnO in the bound state, which indicates the environmental safety of the developed biopolymer film.

## 1. Introduction

Polymer nanomaterials, films, and coatings are widely used in various industries, including pharmaceuticals and medicine. Polymer tapes and films are used in equipment construction as working elements [1,2], to ensure the tightness of threaded joints during the life cycle of the product [3,4], for elastic fixation of broken bones during surgery on human limbs [5,6], for filtering liquids and gases [7,8], for aqueous nanostructures containing antimicrobials of natural origin have been developed to inactivate SARS-CoV-2 surrogates on surfaces [9], for wound healing [10,11,12] and dentistry [13,14,15].

Due to high antimicrobial and antibacterial properties, ZnO NPs are often used in composite materials and films for various purposes. Motelica et al. [16] proposed a simple and affordable method of decontamination to protect paper documents, books, works of art, etc; this method is based on the use of ZnO NPs as antimicrobial agents, and provides a reliable antibacterial and antimicrobial effect. Motelica et al. [17] proposed to use biodegradable alginate-based packaging as a biodegradable polymer instead of petroleum-based polymeric materials. To ensure the antibacterial activity of films, they added different amounts of ZnO NPs and citronella essential oil to find their effect on on two gram-negative (*Escherichia coli* and *Salmonella Typhi*) and two gram-positive (*Bacillus cereus* and *Staphylococcus*). The study showed the prospects of alginate films filled with ZnO and citronella essential oil NPs for cheese preservation. Preventing the possibility of secondary infection of wounds and creating favourable conditions for healing significantly depends on the chosen treatment methods and medical materials used, particularly therapeutic biopolymer films [18,19]. A review by Kumar et al. [20] described the antimicrobial properties of nanomaterials and shows the effectiveness of ZnO NPs in medical practice. Biodegradable composite polymer materials are of considerable interest at the present stage of the development of tissue engineering. The ability to combine modern biotechnology with nanotechnology helps to create new innovative products. Recently, scientists have developed a wide range of biomaterials and new structures and systems based on them for tissue engineering [21]. Bio-based materials have several additional functions, such as unique chemical structure, biological activity, non-toxicity, biocompatibility, biodegradability, recyclability, etc., which position them well in the modern global materials sector and are an alternative to petroleum-based polymers. Prasad [22] investigated bioresorbable polymeric materials for biofilms and other biomedical applications. Song et al. [23] proposed to use a biofilm based on antimicrobial peptides to fight bacteria. Kalia et al. [24] consider biodegradable polymers, especially polyhydroxyalkanoates (degradable biopolymers) as an alternative to traditional polymers. Such polymers are applicable in different ways for agricultural and medical sectors, however, they have low mechanical strength. In addition, the complexity of the production technology and the high cost of the components restrain their wide implementation. It is assumed that biopolymer coatings will reduce the organism’s reaction to a foreign body and thus increase its biocompatibility [25,26]. Infections associated with orthopaedic implants can have catastrophic consequences for patientsA promising direction in preventing surgical infections and accelerating wound healing is the use of therapeutic nano-containing biopolymer films [27,28] To effectively use biopolymer films, it is necessary to carefully study their physical, mechanical and therapeutic properties and ways to correct them. Vargas et al. [29], the application of ultrasound, ohmic heat, and ultraviolet radiation to adapt biopolymer films and coatings to specific application conditions was studied. Croitoru et al. [30] proposed drug delivery to target tissues by external electrical stimulation; they used a new micro-matrix developed on poly (lactic acid) (PLA) and graphene oxide by electrospinning; they loaded quercetin, a natural flavonoid into the fiber matrix to investigate its potential as a modeling agent for dressings. Physicochemical properties and antimicrobial analysis and biocompatibility were also investigated. In order to create effective antimicrobial surfaces, the study [31] proposed to use Bioactive ZnO coatings deposited by the method of matrix pulsed laser evaporation (MAPLE). Dincă et al. [32] presented the effect of a new antibacterial and biocompatible nano-ZnO-bacterial cellulose material with controlled interfaces, formed by MAPLE; this study considered the following microorganisms in vitro (*Escherichia Coli*, *B. subtilis Spizizenii Nakamura*, *Candida albicans*) and mammalian cells (human dermal fibroblast cells) response. The effect of the addition of thymol nanoemulsions on the physical properties and antimicrobial activity of biopolymer films was studied in work [33]. Therapeutic enzymes play an important role in modern medicine, but their use in clinical medicine is very expensive due to their low stability and bioavailability. Immobilization methods are used to improve the effectiveness of therapeutic enzymes [34]. Vasile et al. [35] developed a new composite material for films containing ZnO NPs coated with gentamicin and incorporated into a chitosan matrix to obtain a ZnO/gentamicin-chitosan gel. Excellent antimicrobial properties, growth inhibition of Staphylococcus aureus and Pseudomonas aeruginosa in both planktonic and surface conditions have been established. The results of studies indicate the antibacterial activity of all three components with controlled release of the antibiotic. Spirescu et al. [36] developed biofilms based on ZnO and linalool NPs. In vitro assays for prokaryotic cells have shown that biofilms prevent the formation of both gram-positive and gram-negative bacterial strains. The article [37] describes the synthesis, characteristics and biomedical application of NPs of metal oxides ZnO and Ag_2_O and their combined effect on cell viability. It is established that the nanocomposite can be used as a vehicle for drug delivery systems. Rayyif et al. [38] described the use of ZnO NPs in the composition modified dressings to suppress pathogens of wounds. The works [39,40] studied the antibacterial activity of ZnO NPs on various microorganisms. It proposed a mechanism of action for the model organism of *E. coli* by analyzing the growth, permeability and morphology of bacterial cells. Biopolymer delivery systems based on nanofibers filled with cephradine from gelatin/polyvinyl alcohol, in which the antibiotic is encapsulated in a specialised dosage form have great potential, contributing to high local drug concentrations at the site of infection, controlled drug release, and less drug degradation. Such technologies are already used in the treatment of diabetic wounds [41], for targeted drug delivery [42], to fight bacteria and biofilm in dentistry [43], etc. 

Many studies are devoted to the development of the composition of nanofilled polymer materials [44] and coatings [45,46,47], their application in various fields of technology [48], and the reuse of polymers after processing [49,50,51]. For the rational choice of the composition of nano-containing polymer materials and technological modes of their manufacture [52,53], tests were performed under different types of loads [54]. It was established that the introduction of nanoadditives in the composition of oil composites provides increased wear resistance of friction pairs [55]; moreover, the development and introduction in the medical practice of the composite materials made of polymer and metallic materials with the use of modern additive technologies 3D-Printed were carried out [56,57], including attractive options for creating hierarchical polymer structures on different scales with nanoadditives [58]. Agarwal [59] believes that the combination of renewable biomaterials, fillers, and additive production technologies will significantly improve the physical and mechanical properties of biopolymer films. Today, corrosion of metal products exposed to aggressive environmental influences remains an urgent problem. Examples are prostheses and implants in a living organism’s physiological environment [60,61], underground and underwater structures operated in the natural environment [62,63], etc. One of the widely used approaches to solve the problem is using inhibitors that form an adsorption layer on the surface, thereby blocking the penetration of active substances and reducing the corrosion rate. To overcome these shortcomings, it is proposed to combine natural anti-corrosion substances with biopolymer films or coatings [64,65]. For studying the process of destruction and degradation of metallic materials and thin films of vegetable and mineral oils, research based on electrochemical phenomena is conducted [66,67]. Atomic force microscopy is used to study the fine structure and properties of biopolymers and hydrogels, including silicon probes with functional coatings [68,69]. Thus far, significant progress has been made in the mechanics of thin films, coatings, and overlays. In recent years, analytical and numerical studies of the stress-strain state of composite elastic materials and coatings with various applications have been carried out [70]. Injection technologies have excellent prospects for treating surface defects using malleable polymer compositions [71]. In Ref. [72], the dependences for determining the stress-strain state of the layer of the elastic polymer material of the belt caused by shear were established, and the strength conditions for the structure were formulated. In Ref. [73], the stress-strain state and the process of destroying the biofilm by air-jet were studied. Computational thermodynamics methods are currently widely used to model the properties of composite materials and coatings based on them [74,75,76], and calculations from the first principles within the theory of electron density functional are applied [77,78]. Myronyuk et al. [79] used a mathematical approach to describe and analyse the characteristics of polymer materials and investigated the effect of structure on the wear resistance of superhydrophobic coatings. Horn et al. [80] presented the results of modelling biofilms, which are considered viscoelastic systems that can grow and develop and deform or even shift under the action of external forces; however, such approaches are difficult to adapt for polymer nano-containing materials. Rakhmanova et al. [81] is devoted to applying response surface methodology for optimization of nanosized ZnO synthesis conditions by electrospinning technique. 

An analysis of the results of studies, in particular system showed that biopolymer films containing ZnO NPs are technologically advanced, have antibacterial activity, and are promising for use as hydrophilic bandages in the treatment of wounds and burns; however, there is almost no information about the influence of component composition and technological modes of formation of biopolymer nanofilled films on their physical, mechanical, and therapeutic properties, which complicates the process of their development and effective use in medical practice for patients; however, a restraining factor in biopolymer films and coatings is a certain instability of their physical, mechanical, and barrier properties. Therefore, during the development of polymer materials, it is necessary to conduct many experiments on the choice of their rational component composition and technological modes of forming biopolymer nanofilled films. To reduce the number of experiments during technology development, the response surface methodology for optimization is used [81,82,83].

Purpose and tasks of research.

The work aims to build a mathematical model of the process of forming therapeutic biopolymer nanofilled films and establish optimal values of technological parameters and component composition to ensure the required therapeutic effect with the high physical and mechanical properties of the film.

To achieve this goal, the following tasks were set:to build a mathematical model of the process of formation of therapeutic biopolymer nanofilled films;to study the influence of component composition and technological parameters of the process of formation of therapeutic biopolymer nanofilled films on their therapeutic and physical-mechanical properties;to choose the optimal component composition and technological parameters for forming therapeutic biopolymer nanofilled films, which provide the maximum therapeutic effect with sufficient vapor permeability, biodegradation duration, swelling, and elasticity;conduct a microscopic study of ZnO NPs and biopolymer nanofilled films;to study antimicrobial properties and biopolymer nanofilled films.

## 2. Materials and Methods

### 2.1. Research Materials and Equipment

We choose the following components for biopolymer nanofilled films based on literature review and the experience of our own previous studies: gelatin, polyvinyl alcohol, distilled water, lactic acid, glycerin and zinc nanooxide.

To form prototypes of biopolymer nanofilled films, we used chemically pure components produced by gelatin—Weishardt International, Liptovský Mikuláš, Slovakia; polyvinyl alcohol—Sure Chemical Co., LTD. Shijiazhuang, China; lactic acid solution (80%)—Cosco, Shanghai, China; Glycerin—VG, Gerimpeks, Rellingen, Germany; ZnO nanopowder (20–30 nm)—Hongwu new material, Guangzhou, China. We used Aquadistillator DE-4-2 to obtain distilled water.

The proposed wound-healing biopolymer nanofilled films were obtained according to the developed technological process (Figure 1).

We weighed the components used AXIS AD200 balance with an accuracy of 0.001 g. We took the original composition of the film according to the patent UA110594U [84]. We varied only the ratio of the amounts of gelatin to polyvinyl alcohol and the mass fraction of zinc nanooxide, and the mass fractions of the other components remained unchanged; we also varied temperature and exposure time during the experiments according to the plan. We heated the mixture in a microwave oven SAMSUNG GE88SUB/UA (maximum power of ultra-high frequency radiation 800 W). 

At the first stage of preparation of the mixture, gelatin, polyvinyl alcohol and water were added to the fluoroplastic beaker and stirred with a glass rod. The resulting mixture was heated to a given temperature at a high-frequency radiation power of 300 W, stirring it periodically every 30 s.

At the second stage, stirring the mixture, alternately a solution of lactic acid (80%) and glycerin was added and again heated to a given temperature at a power of ultra-high frequency radiation of 300 W.

At the third stage, stirring the mixture, nanopowder of ZnO was added and subsequently kept in a microwave oven at a power of 150 W for an appropriate period of time according to the experimental plan.

Next, the mixture was poured into a mold with a chrome coating and kept to complete the polymerization process for 24 h at room temperature. The formed film was removed from the mold, and its quality was visually inspected using a seven-fold Attache MG1004A magnifier.

To ensure the quality of the working surfaces of the molds for the manufacture of samples of polymer composite nanofilled materials applied chrome coatings [85], and during the formation of samples considered, the damage of the material [86].

### 2.2. Methods for Studying the Properties of Biopolymer Nanofilled Films

#### 2.2.1. Elasticity

Under in vitro conditions, the elongation of samples after breaking was determined by the results of tensile strength tests on a testing machine type FPZ-10/1 in the «Laboratories of certification tests of anti-corrosion insulating coatings of pipelines» based on Karpenko Physico-mechanical Institute of the National Academy of Sciences of Ukraine, Lviv. The film’s elasticity of wound-healing biopolymer nanofilled films was determined by testing the film samples under uniaxial stretching at a temperature of 19 °C and an air humidity of 80%, according to State Standart GOST 14236-81.
(1)$$E=\frac{\Delta l_{0r}}{l_{0}}\cdot 100\%\;,\;$$
where $\Delta l_{0r}$ is the change in the estimated length of the sample at the moment of breaking, mm, $l_{0}$ is the initial estimated length of the sample, mm. 

The geometric dimensions of the samples were changed using a calliper with an accuracy (±0.1 mm).

#### 2.2.2. Vapor Permeability

Vapour permeability studies were performed by gravimetry at the Department of Biological and Medical Chemistry, Ivano-Frankivsk National Medical University. The polymer film was hermetically fixed on a plastic cup 40 mm in diameter filled with distilled water and weighed on an electronic balance after 24 h. The study was performed three times on a thermostat at a temperature of 37 °C.

The formula calculated the vapour permeability of the films:(2)$$P=\frac{m_{1}-m_{2}}{S\cdot t}\;,$$
where *m*_1_ is an initial mass of the structure with water, g; *m*_2_*—*a final mass of the structure with water, g; *S* is the actual area of the film, cm^2^; *t—*research time, h.

#### 2.2.3. Degradation Time

The degradation of the material was studied at the Department of Biological and Medical Chemistry, Ivano-Frankivsk National Medical University, by placing the polymer film in a plastic cup filled with distilled water at a temperature of 37 °C. Every hour, the film was removed, and its structure was visually observed. The film was considered to degrade when it could not be removed from the cup and did not retain its previous structure and shape.

#### 2.2.4. Isolation of ZnO NPs

The study of the isolation of ZnO NPs from a polymer film was carried out using a calorimetric test system for zinc Aquaquant^®^ company Merck KGaA (Darmstadt, Germany) with a sensitivity of 0.1–5 mg/L, under the conditions of obtaining the film by the method of formation of Zn^2+^ complexes with film components (gelatin, olivinyl chloride, lactic acid. acid*—*known as complexones).

Samples of the films were placed in chemical beakers of distilled water, and samples of the solution were taken after 5, 15, 30, 60 min, and 24 h, respectively. Detection of Zn^2+^ was performed on a colored blue-green complex formed by the reaction of the selected solution with thiocyanate ions in the presence of diamond green. The calibration graph was based on samples with well-known values of ZnO content in the film.

The concentration of zinc ions was determined on a scale of colour intensity, which corresponded to specific concentrations. Based on the obtained results of zinc concentration, ZnO evolution was determined taking into account the known ratio ZnO/Zn = 81.408/65.38.

We took the number of selected ZnO NPs for 24 h in the calculations. 

#### 2.2.5. Swelling

According to the generally accepted method, the study of the degree of swelling of biodegradable polymer material was conducted at the Department of Biological and Medical Chemistry, Ivano-Frankivsk National Medical University. The samples were weighed on an electronic balance and immersed in distilled water at 37 °C. After 24 h, the samples were removed from the water, the residual liquid was removed with filter paper, and the film was weighed again.

The degree of swelling was calculated by the formula
(3)$$S_{w}=\frac{m_{1}-m_{0}}{m_{0}}\;,$$
where $m_{1}$*—*the mass of the film after exposure to water; $m_{0}$*—*the mass of the film before immersion in water.

#### 2.2.6. Antibacterial Activity

Antibacterial activity of wound-healing biodegradable polymer material was evaluated in the bacteriological laboratory of the Department of Microbiology, Ivano-Frankivsk National Medical University, using the disc diffusion method. From polymer films, which are saturated with the active substance ZnO NPs, disks with a diameter of 6 mm were cut out and tested for antimicrobial activity against a series of clinical strains of opportunistic pathogens: *Staphyloccus aureus* (*S. aureus*), *Streptococcus pyogenes* (*S. pyogenes*), *Escherichia coli* (*E. coli*), *Pseudomonas aeruginosa* (*P. aeruginosa*), *Candida tropicalis* (*C. tropicalis*). The value of the diameter of the zone of growth retardation (diameter of action) of microorganisms of the studied disks of polymer material was determined by processing photographs using the computer program UTHSCSA ImageTool 2.0.

#### 2.2.7. Observation of the Microstructure

We conducted the microscopic studies of ZnO NPs and biopolymer nanofilled films at the Center for collective use of scientific instruments “Center for Electron Microscopy and X-Ray Microanalysis” of the Karpenko Physico-mechanical Institute of the National Academy of Sciences of Ukraine, Lviv. We used a “Stemi 2000-C” optical microscope, a ZEISS EVO 40XVP scanning electron microscope with a micro X-ray spectral analysis system, and an INCA ENERGY 350 energy dispersive X-ray spectrometer. Before conducting electron microscopic studies, we applied a monolayer of gold to the surface of the samples to increase their conductivity. We studied the chemical composition of the phases by energy dispersive X-ray spectroscopy (EDS).

### 2.3. Methods of Planning Experimental Research

An orthogonal central compositional plan (OCCP), which is the simplest second-order plan, was chosen to construct the experimental design; this plan allows us to obtain regression models of the second order of elasticity *Y_E_*, vapour permeability *Y_P_*, degradation time *Y_td_*, isolation of ZnO NPs *Y_V_* and swelling *Y_H_* from the component composition and technological parameters of its manufacture (concentration ratio *C*, the content of ZnO NPs *I*, exposure time in the furnace *t* and heating temperature *T*). Recommendations [87] were used when choosing parameters.

Regression dependence was sought in the form:*Y_E_* = *f* (*C*, *I*, *t*, *T*);  *Y_P_* = *f* (*C*, *I*, *t*, *T*);  *Y_td_* = *f* (*C*, *I*, *t*, *T*);*Y_V_* = *f* (*C*, *I*, *t*, *T*);  *Y_H_* = *f* (*C*, *I*, *t*, *T*);  *Y_E_* = *f* (*C*, *I*, *t*, *T*).(4)

We also chose this plan because it is not the regression coefficients that are more important when estimating the outer region, but the response function itself. In addition, it allows us to simplify the regression model by rotating the coordinate axes significantly, that is, by transforming the coordinates. Orthogonal planning, which provides an error in predicting the initial value of the regression equation, depends only on the distance of the point of factor space to the centre of the experiment. It allows us to predict the value of the response function with equal accuracy and, consequently, to transform the coordinate system to simplify the regression equation.

Orthogonal plans are optimal because they minimise systematic errors associated with the inadequacy of the study results of second-order polynomials.

The total number of experiments in an orthogonal central composition experiment depends on a number of factors. The choice of the number and coordinates of “star” points must meet such requirements:(a)the total number of experiments significantly exceeds the number of regression coefficients;(b)factors change on a small number of levels.

In this type of experiment, the experiments are performed on a matrix containing three groups of experiments, which are placed symmetrically and at the same distance from the centre of the experiment: points of complete or fractional factorial experiment *N* = 2*^n^*;“star” points (plan type “cross”) *N*_α_ = 2*n*;zero (central) points *N*_0_.

The choice of the size of the star shoulder *α* and the number of experiments in the centre of the plan *N*_0_ is related to the optimality criterion of the plan.

The formula determines the total number of *N* points of the OCCP
$$N=N_{\varphi }+N_{\alpha }+N_{0}.$$

That is
(5)$$N=2^{k}+2k+n_{0},$$
where *k**—*a number of factors; $n_{0}$*—*a number of experiments in the centre of the experiment plan (usually one or two experiments).

The experiments were conducted at five coded levels (−*α*, −1, 0, +1, +*α*). The interval of change of factors should be such that the change range covers a stationary area of factor space (Appendix A, Appendix A).

The limits of variation of the component composition and technological parameters of the process of obtaining the film material were chosen for the following reasons:when the ratio of the concentrations of the main components of the film, gelatin (G) to vinyl alcohol (V), are less than 1.2 is obtained very rapid degradation of the film, and at a ratio of more than 3.2 the film is characterised by high hardness and does not have good elastic properties;at the content of ZnO of less than 50 mg/L, the therapeutic (antimicrobial action) is not provided, and at the content of more than 230 mg/L, the therapeutic efficiency (antimicrobial action) of a film practically does not increase anymore;the exposure time in the furnace, t, min was chosen based on the need for complete (through) uniform heating of the mixture of film components in the crucible;the heating temperature *T*, °C of the mixture of film components were taken to ensure complete softening and dissolution of the components in water and ensure a uniform consistency of the film material.

The OCCP was conducted at five levels of factor variation. At star points, considering four factors, the size of the shoulder of the experiment is *α* = 1.4826, and at the centre of the experiment. The selected factors meet all the requirements for them. The accuracy of maintaining the technological parameters of obtaining material was 3–5%. The intervals of variation are given in Table 1.

To exclude the influence of unregulated and uncontrolled factors during the experiments on the values of the optimization parameters, randomisation of the planning matrix was performed by the method of random balance, implemented by a random number generator. 

A second-order polynomial was used to describe the technological process of obtaining the material (four factors)
(6)$$Y=b_{0}+\sum_{i=1}^{k}b_{i}x_{i}+\sum_{i\neq j}^{k}b_{ij}x_{i}x_{j}+\sum_{i=1}^{k}b_{ii}x_{i}^{2},$$
where $b_{0},\;\;b_{i},\;\;b_{ij},\;\;b_{ii}—\mathrm{regression}\;\mathrm{coefficients},\;x_{i}—\mathrm{factors}\;\mathrm{of}\;\mathrm{the}\;\mathrm{experiment}$.

The total number of regression coefficients in polynomial (9) was determined from the expression
(7)$$M_{k}=\frac{1}{2}\left(M+1\right)\left(M+2\right),\;$$
where *M*—number of experimental factors.

For four variables, the number of regression coefficients is $M_{k}=15$.

The coefficients of the approximating polynomial (6) and their variances were calculated according to known formulas. The significance of the regression coefficients was assessed by the Student’s test. The value of the t-criterion (also called observational or actual) found from observational data was compared with the tabular (critical) value determined from Student’s distribution tables.

The tabular value is determined depending on the level of significance (α) and the number of degrees of freedom, which in the case of linear pairwise regression is equal to (*n* − 2), *n*—number of observations. Thus, the tabular value of t-statistics depends on the confidence probability, the number of factors, and the original series’s length.

Suppose the actual value of the t-criterion is greater than the tabular one (modulo). In that case, the main hypothesis is rejected, and it is assumed that with probability (1 − α), the parameter or statistical characteristic in the general population differs significantly from zero. 

Suppose the actual value of the t-criterion is less than the tabular (modulo). In that case, there is no reason to reject the main hypothesis, i.e., the parameter or statistical characteristic in the general population of insignificant differs from zero at the level of significance α.

To find the optimal values of the technological parameters of film formation, we determined the partial derivatives of the response functions. We equated them to zero, obtaining a system of equations. The solution allowed us to determine the optimal values of the technological parameters of the process.

Based on the results of calculations performed on a PC using Statistica software developed by StatSoft, the response surfaces of the optimization parameter and their two-dimensional cross-sections were built. The optimal values of the component composition and technological parameters of obtaining a biopolymer nanofilled films were also determined. 

## 3. Results

### 3.1. Optimization of Biopolymer Nanofilled Film Composition and Their Properties

To substantiate the technological parameters of the process of creating a wound-healing biopolymer nanofilled films and to establish patterns of changes in elasticity *Y_E_*, vapor permeability *Y_P_*, degradation time *Y_td_*, isolation of ZnO NPs *Y_V_* and swelling *Y_H_* of the obtained film from the component composition and technological parameters of its manufacture (concentration ratio *C*, the content of ZnO NPs *I*, exposure time in the furnace t and heating temperature *T*) laboratory experimental studies of samples of polymer film created by the developed technology was conducted. To evaluate the quality of the film, its elasticity *Y_E_*, vapour permeability *Y_P_*, degradation time *Y_td_*, isolation of ZnO NPs *Y_V_* and swelling *Y_H_* were measured.

Unknown coefficients of the regression equation were determined by the matrix method in coded form. The assessment of the significance of the regression equation coefficients was checked at a level of 0.05 using Student’s *t*-criterion. Statistically insignificant coefficients were rejected. After that, the significant coefficients of the second-order regression equations were substituted into the formula (6). 

Response functions (optimization parameter), which reflect the dependence of elasticity *Y_E_*, vapor permeability *Y_P_*, degradation time *Y_td_*, isolation of ZnO NPs Y_V_ and swelling *Y_H_* of the obtained film depending on the ratio of concentration *C*, the content of ZnO NPs *I*, exposure time in the furnace *t* and heating temperature *T* according to the results of the OCCP of the experiment in the natural values of the variable factors took the following form for:

– elasticity
(8)$$Y_{E}=89.661+2.370C^{2}-0.268I+0.001I^{2}+0.607T-0.005T^{2}+\;+0.046CT-0.001IT,$$

– vapor permeability
(9)$$Y_{p}=784.858+217.603C+52.146C^{2}+0.826I-0.002I^{2}-30.434t+\;+2.921t^{2}-4.555T+0.046T^{2}-0.006IT,$$

– time of degradation
(10)$$Y_{td}=33.210-10.076C+3.996C^{2}+0.167I-0.0002I^{2}+0.221t^{2}+\;+0.004T^{2}-0.017CI-0.402Ct-0.001IT-0.022tT,$$

– isolation of ZnO NPs
(11)$$Y_{V}=22.116-11.138C+4.436C^{2}+1.269I-0.003I^{2}+0.439t^{2}-0.498T+\;+0.007T^{2}-0.032CI-0.098CT-0.024It-0.001IT,$$

– swelling
(12)$$Y_{H}=449.109+12.566C^{2}-1.026I+0.002I^{2}-13.580t+1.356t^{2}+\;+0.015T^{2}+0.116CI-0.459CT-0.079tT.$$

According to the obtained response functions (8)–(12), the graphic dependences of elasticity *Y_E_* (Figure A1), vapor permeability *Y_P_* (Figure 2), degradation time *Y_td_* (Figure 3), isolation of ZnO NPs *Y_V_* (Figure 4) and swelling *Y_H_* (Figure A2) of the obtained film on the concentration ratio *C*, the content of ZnO NPs *I*, exposure time in the furnace t and heating temperature *T* were plotted.

Analysing the results of studying the elasticity of the film on the composition and technological parameters of its manufacture (Figure A1a–c), we see that an increase in the ratio of the concentrations of the main components of the film to 1.6 causes a decrease in its elasticity. A further increase in this parameter leads to increased elasticity. When the content of ZnO NPs increases from 40 mg/L to 170 mg/L, the elasticity of the biopolymer film decreases (Figure A1a,d,e). There is a zone of minimum elasticity within the concentration of ZnO NPs from 160 mg/L to 180 mg/L. A further increase in the concentration of ZnO NPs leads to a slight increase in the elasticity of the biopolymer film. The study of the influence of the exposure time of the components of the mixture under heating (Figure A1b,d,f) on the change in the elasticity parameter of the biopolymer film showed that the duration of the heating time does not strongly affect the change in the elasticity of the biopolymer film. With increasing exposure time at temperature, it decreases. The studies of the effect of the heating temperature of the mixture on the elasticity (Figure A1c,e,f) found that within the temperature change from 20 °C to 50 °C, the elasticity of the biopolymer film increases, and a further increase in the heating temperature of the mixture leads to a decrease in the elasticity parameter.

Analysing the results of studies of the vapour permeability of the film on the composition and technological parameters of its manufacture (Figure 2a,b,c), we see that an increase in the ratio of the concentrations of the main components of the film to 2.2 causes a decrease in its vapour permeability. A further increase in this parameter leads to its growth. With increasing content of ZnO NPs from 40 mg/L to 120 mg/L, vapour permeability of the biopolymer film increases (Figure 2a,d,e), and within the concentration of ZnO NPs from 115 mg/L to 125 mg/L, there is a zone of maximum vapour permeability. A further increase in the concentration of ZnO NPs leads to a decrease in the vapour permeability of the biopolymer film due to the compaction of the structure of the gel film by NPs. The study of the effect of exposure time of the components of the mixture under heating (Figure 2b,d,f) on the change in the vapour permeability parameter of the biopolymer film showed that with increasing heating time, it decreases and has a minimum in the range from 6.5 min to 7.5 min. With a further increase in the exposure time of the mixture’s components, the vapour permeability of the biopolymer film increases insignificantly. The studies of the effect of the heating temperature of the mixture on vapour permeability (Figure 2c,e,f) found that a change in temperature in the range from 20 °C to 60 °C leads to a decrease in vapour permeability. At a temperature of 60 °C, there is a minimum of the studied parameter. A further increase in the heating temperature of the mixture leads to an increase in the vapour permeability parameter.

The influence of the ratio of the concentrations of the main components of the biopolymer film on the time of its degradation has been studied (Figure 3a–c). Analysis of the results showed that when the ratio of the components of the mixture to 1.9, there is a slight decrease in the degradation time of the biopolymer film, and a further increase in the ratio of the concentrations of the components of the mixture increases the time of degradation. In the analysis of experimental results of the study of the effect of ZnO NPs on the degradation time of the biopolymer film (Figure 3a,d,e), it was found that increasing the concentration of nanooxide increases the degradation time of the film. Analysis of the results of the study of the exposure time of the components of the mixture at a given heating temperature showed that changing the exposure time at a given temperature from 1.0 min to 3.0 min contributes to a slight reduction in degradation time (Figure 3b,d,f), in the range from 3.0 min to 3.5 min the minimum of this parameter is observed. The reduction of the degradation time in the first minutes of heating the mixture is associated with the gelatin transition from gel-like to liquid. A further increase in the exposure time in the furnace leads to an increase in the parameter of the degradation time of the film due to the crystallisation of polyvinyl alcohol molecules and the re-gelation of the mixture. The study of the influence of the heating temperature of the mixture on the degradation time of the biopolymer film (Figure 3c,e,f) showed that with increasing heating temperature of the mixture from 20 °C to 30 °C the degradation time of the biopolymer film increases slightly. At a temperature from 30 °C to 35 °C, the minimum of this parameter is observed. A further increase in the heating temperature leads to a more significant increase in the degradation time of the biopolymer film due to the formation of a stable structure of the film caused by the crystallisation of polyvinyl alcohol.

Figure 4a–c shows that changing the ratio of the concentration of the main components of the mixture to manufacture biopolymer film does not significantly affect the isolation of ZnO NPs from it. In this case, increasing the ratio of concentrations to 2.4 leads to a decrease in the isolation of ZnO NPs from the film. In the range from 2.4 to 2.8, there is a minimum of its isolation, and a further increase in the concentration ratio leads to an increase in its isolation. With an increase in the content of ZnO NPs in the biopolymer film (Figure 4a,d,e) to 200 mg/L, an increase in its isolation is observed. Still, when this value is reached, a further increase in it leads to a meager improvement in the isolation of these NPs from the film. Increasing the exposure time of the components of the mixture at a given temperature during the preparation of the biopolymer film (Figure 4b,d,f) initially leads to a decrease in the isolation of ZnO NPs from the film, the minimum of which is obtained at 4.8 min exposure. A further increase in the exposure time of the mixture’s components causes an increase in the isolation of ZnO NPs. The heating temperature of the components of the mixture of biopolymer film (Figure 4c,e,f) affects the amount of the isolation of ZnO NPs from the film in this way: with increasing heating temperature, the isolation of ZnO NPs from the film decreases, and within the temperature from 55 °C to 65 °C, its minimum value is observed; further increase in the temperature of the heating mixture does not cause an increase in the isolation of ZnO NPs from the biopolymer film.

Analysing the results of studies of film swelling from the composition and technological parameters of its manufacture (Figure A2a–c), we see that increasing the concentration of the main components of the film to 1.4 causes a decrease in its swelling, and further increase of this parameter leads to increased swelling. With increasing content of ZnO NPs from 40 mg/L to 180 mg/L, the swelling of the biopolymer film decreases (Figure A2a,d,e), and within the concentration of ZnO NPs from 180 mg/L to 200 mg/L, there is a zone of its minimum. A further increase in the concentration of ZnO NPs leads to a slight increase in the swelling of the biopolymer film. The effect of exposure time of the components of the mixture under heating (Figure A2b,d,f) on the change in the swelling parameter of the biopolymer film showed that increasing exposure time at temperatures up to 6 min swelling decreases. A further increase in exposure time at temperature leads to an increase in the swelling parameter. The studies of the effect of the heating temperature of the mixture on the swelling (Figure A2c,e,f) found that within the temperature change from 20 °C to 50 °C, swelling of the biopolymer film decreases, and a further increase in the heating temperature of the mixture leads to an increase in the swelling parameter.

To find the optimal values of the technological parameters of the film manufacture, we determined the partial derivatives of the response functions (8)−(12), equated them to zero, and obtained a system of four equations. The solution of the system of equations allowed to determine the optimal values of technological parameters of the process (Table 2), which provide the manufacture of biopolymer nanofilled films with maximum healing ability and sufficiently high physical and mechanical properties: elasticity, vapour permeability, degradation time and swelling.

As far as the therapeutic (wound-healing) properties of the biopolymer nanofilled films mainly depend on the release of ZnO NPs and are responsible for the diameter of action—The destruction of bacteria, we took the technological options and the optimal component composition, which provide the maximum isolation of ZnO NPs *Y_V_* = 114.4 mg/L. After that, the predicted values of the corresponding optimization parameters were calculated from the obtained regression Equations (8)−(12). The results of the calculation are summarised in Table 3.

### 3.2. Microscopic Studies

We formed biopolymer nanofilled films (Figure 5) according to the obtained optimal values of technological options.

According to Figure 5, biopolymer nanofilled films have a uniform texture and do not have visible defects such as pores, cracks, caverns and others.

Figure 6 shows the results of studying the morphology and composition of ZnO NPs by the SEM/EDS methods.

In the study, we analyzed the SEM/EDS micrographs of ZnO NPs, which are shown in Figure 6. It was established that the weight percentage of zinc was 88.2, and that of oxygen was 11.8, which is quite close to the stoichiometric ratio of the components in the compound with the formula unit (formula unit) ZnO. The distribution of Zn and O is quite uniform (Figure 6c,d).

The microstructure and composition of biopolymer nanofilled films were also investigated by the SEM/EDS method (Figure 7).

The results of SEM/EDS analysis of the microstructure of the film section (Figure 7) indicate the presence of a uniform distribution of ZnO NPs. Analysis of the spectrum (Figure 7b) shows that the content of the main elements in the film is as follows (Wt., %); Zn—16.67; O—28.81; C—54.51. The intensity of carbon is highest in the zones corresponding to the matrix of the film (Figure 7c). Traces of Zn (Figure 7d) and O (Figure 7e) coincide in the nature of their location on the distribution maps (EDS maps) (Figure 7), which indicates the presence in these zones of a film of the ZnO compound, which has an antibacterial effect effect on microorganisms.

### 3.3. Antimicrobial Action

The biopolymer nanofilled films with optimal composition of the active substance ZnO NP, showed high antimicrobial activity against a number of clinical strains of opportunistic microorganisms: *S. aureus*, *S. pyogenes*, *E. coli*, *P. aeruginosa*, *C. tropicalis.*


Figure 8 presents the diameter of the protective effect of biopolymer nanofilled films, manufactured according to the optimal composition and technological parameters, for the studied bacteria.

According to the results of microbiological studies as shows the diagram (Figure 7), biopolymer nanofilled films with antibacterial activity were tested against both gram-positive (*S. aureus*, *S. pyogenes*), gram-negative (*E. coli*, *P. aeruginosa*) and fungi (*C. tropicalis*). The diameter of the protective action of biopolymer nanofilled films is the largest when it affects *S. pyogenes*, since it is the most sensitive to the active substance while the rest of the studied bacteria (*S. aureus*, *E. coli*, *P. aeruginosa*, *C. tropicalis*) are also carried out a significant destructive effect is caused by the ZnO NPs. 

Padmavathy et al. [39] described in details the ZnO NPs–microbes interaction using a physio-biochemical analysis. According to the interaction mechanism, ZnO NPs are adsorbed on the surface of bacteria and internalized in the periplasm. ZnO NPs penetrates into bacterial cells, damages the structure of the bacterial cell membrane and inhibits the activity of some membrane enzymes through the formation of reactive oxygen species, which causes the death of bacteria.

## 4. Discussion

Based on the constructed mathematical models, our study analysed the effect of the component composition and technology of forming biopolymer films with nanofillers on their therapeutic, physical, and mechanical properties. Taking as the optimal values of the technological parameters of the process that ensure the manufacturing of biopolymer nanofilled films with the maximum therapeutic ability, the physicomechanical properties of these films were calculated. The optimal values of the technological parameters of forming biopolymer composites containing ZnO NPs are within the factor space. The research results showed that the healing properties of biopolymer films mainly depend on the content of ZnO NPs. Degradation of these biopolymer films provides dosed drug delivery to the affected area. 

Table 3 shows that the most significant deviation of the optimization parameter reaches the swelling *Y_H_*, 21.5%; however, it is insignificant because the most crucial characteristic of the drug is its healing ability—Isolation of ZnO NPs *Y_V_*. For research, the selected system of biopolymer film is formed from available and widely used components, high-tech in manufacturing, provides easy release of the active substance ZnO NPs, convenient for use in medical practice, is biodegradable and does not pollute the environment; this article optimizes the technology of obtaining biopolymer nanofilled films, which allowed to improve its physical, mechanical and therapeutic properties compared to the results of previous studies [19].

Regarding the effect and usefulness of each component of biopolymer nanofilled films: gelatin—A water-soluble crosslinking and polymerizing agent of polyvinyl alcohol; polyvinyl alcohol—Water-soluble polymer of linear structure; distilled water—Solvent; lactic acid—A crosslinking component; glycerin—Emollient component; zinc nanooxide is the active substance. 

The obtained results of the ZnO NPs content influence on inhibitory ability of the developed biopolymer films correlate with the results obtained by other scientists [87,88,89,90].

The articles [39,40] studied the antibacterial activity of ZnO NPs on various microorganisms. The mechanism of action of ZnO NPs on bacteria is described in detail in a study Padmavathy et al. [39], according to which ZnO NPs damages the structure of bacterial cell membrane and depressed the activity of some membranous enzymes by reactive oxygen species production which caused bacteria to die eventually. 

The introduction of innovative technologies with nanomaterials to develop biopolymer films will expand their application in medicine. Therapeutic films, known for their properties as long-acting drugs, have advantages over traditional methods of drug administration; they are easy to use, can be used independently by the patient, and allow for the combination of different groups of active ingredients. Chursin et al. [91] presented technologies for obtaining multicomponent biopolymer composites and films with the optimal polyacrylic acid and alkaline collagen solution ratio. The influence of polysaccharides (sodium alginate and pectin) on the rheological characteristics and properties of biopolymer composite films was analysed.

Lal et al. [14] have found that combined NPs-based agents provide a more favourable release of drugs in a controlled manner and efficient delivery to the site of action. We hope that research will also contribute to the development of biopolymer nanofilled films used in dentistry, given their physicochemical properties, bioactivity and versatility.

In addition, biopolymer films containing ZnO NPs can also be used for food packaging [17].

Work is underway to replace traditional polymers/plastics made from petroleum-based with alternative biodegradable materials. To consider these bioproducts as more viable options than traditional polymer materials, it is necessary to ensure that they are wholly biodegradable or compostable and reduce the release of hazardous compounds into the environment during their degradation [92,93,94].

The probabilistic simulations [95,96] show the need to study the environmental risks of nanomaterials entering the environment, particularly by considering the effects of combinations of different oxides in water and soil.

Giaveri et al. [97] developed a circular economy concept for protein processing that is inspired by nature. In this study, mixtures of several peptides and/or proteins are depolymerized into their amino acid components, and then the resulting amino acids serve as the basis for the synthesis of new fluorescent and bioactive proteins. These proteins may be important in the engineering of medical materials.

The destruction products of the therapeutic biopolymer film developed by us are carbon dioxide, water, and a small amount of ZnO NPs in the bound state, which indicates its environmental safety.

## 5. Conclusions

According to the results of research on the influence of component composition and technological parameters of the process of forming biopolymer nanofilled films:For the first time multifactor mathematical models of the second order for determination of physical-mechanical and medical properties are constructed (the elasticity *Y_E_*, vapor permeability *Y_P_*, degradation time *Y_td_*, isolation of ZnO NPs *Y_V_* and swelling *Y_H_*) on the concentration ratio gelatin to vinyl alcohol *C*, the content of ZnO NPs *I*, exposure time in the furnace *t* and heating temperature *T* were built;Based on the obtained mathematical models, the values of the parameters of the process of formation biopolymer nanofilled films (component composition and technological parameters) were established to ensure optimal characteristics of the polymer film (the elasticity *Y_E_*, vapor permeability *Y_P_*, degradation time *Y_td_*, isolation of ZnO NPs *Y_V_* and swelling *Y_H_*) (Table 2);It was substantiated that to ensure high therapeutic properties of the film, the final optimization must be carried out on the maximum isolation of ZnO NPs, which is provided for such optimal component composition and technological parameters of the process: concentration ratio *C* = 3.2, the content of ZnO NPs *I* = 178.3 mg/L, exposure time in the furnace *t* = 7.1 min and heating temperature *T* = 72.8 °C. And at the same time, high physical and mechanical indicators are provided.It was shown that the calculated values of the optimization parameters: *Y_E_*, *Y_P_*, *Y_td_*, *Y_H_* for the optimal values of the technological parameters of the film at which the maximum*Y_V_* is provided differ by 17.2; 12.8; 21.3; 21.5% respectively.The results of SEM/EDS analysis of the microstructure of a section of biopolymer nanofilled films indicate the presence of a uniform distribution in the matrix of ZnO NPs, which have an antibacterial effect on pathogenic microorganisms.Research results showed high antimicrobial properties due to the release of ZnO NPs from the composition biopolymer nanofilled films.

Further studies are planned to study the effect of various metal oxide NPs on the therapeutic properties of biodegradable polymer materials.

## Figures and Tables

**Figure 1 nanomaterials-12-02413-f001:**
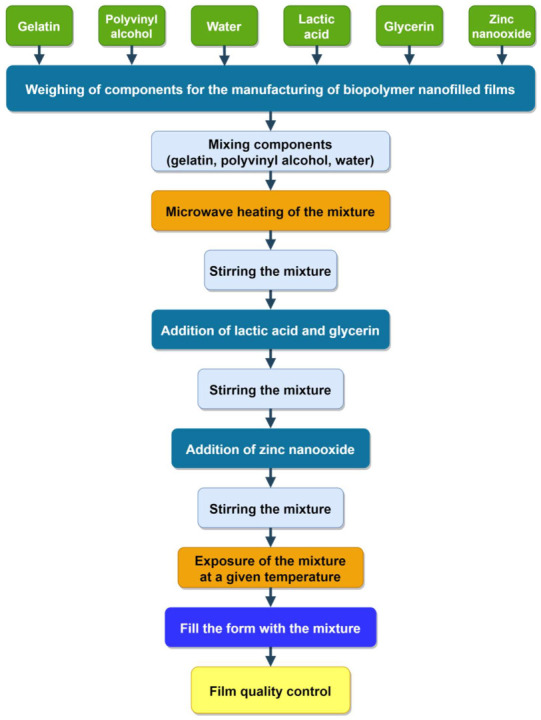
Block diagram of the technological process of formation of biopolymer nanofilled films.

**Figure 2 nanomaterials-12-02413-f002:**
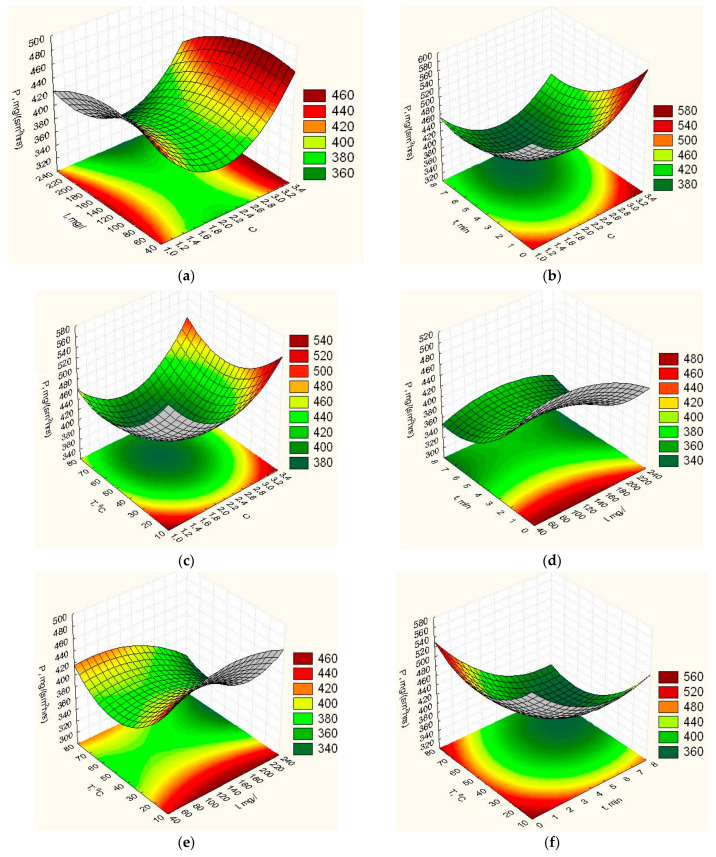
Response surfaces of dependence of the vapor permeability of the film on the composition and technological parameters of its manufacture: (**a**)—*C*, *I*, *t* = 4 min, *T* = 46 °C; (**b**)—*C*, *I* = 140 mg/L, *t*, *T* = 46 °C; (**c**)—*C*, *I* = 140 mg/L, *t* = 4 min, *T*; (**d**)—*C* = 2.2, *I*, *t*, *T* = 46 *°*C; (**e**)—*C* = 2.2, *I*, *t* = 4 min, *T*; (**f**)—*C* = 2.2, *I* = 140 mg/L, *t*, *T*.

**Figure 3 nanomaterials-12-02413-f003:**
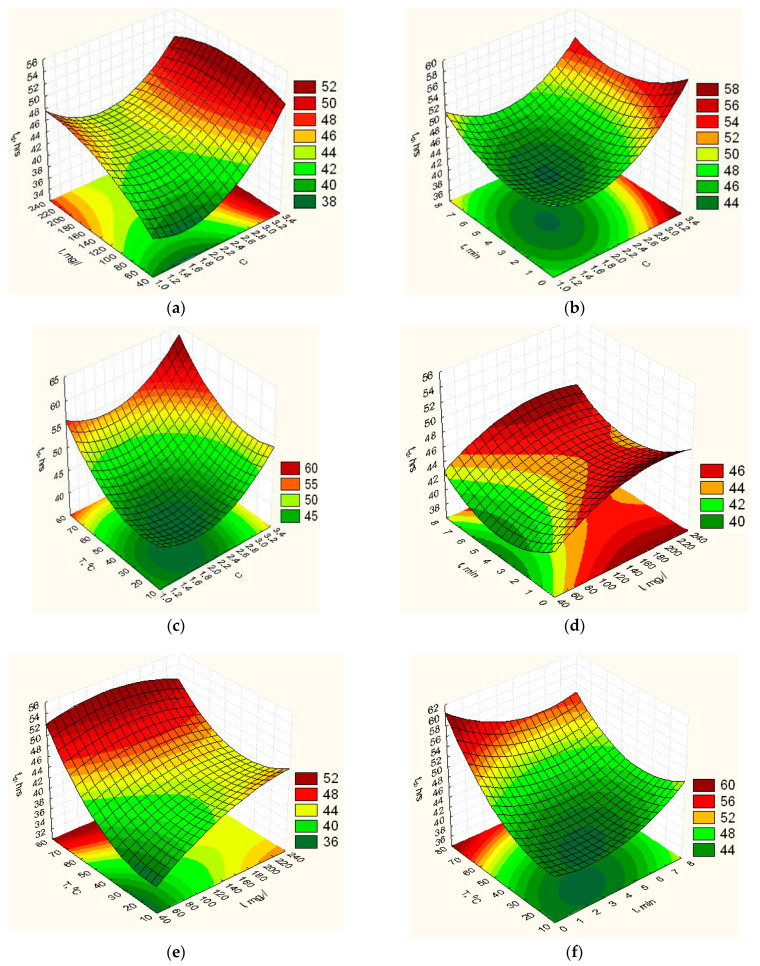
Response surfaces of the dependence of the degradation time of the film on the composition and technological parameters of its manufacture: (**a**)—*C*, *I*, *t* = 4 min, *T* = 46 °C; (**b**)—*C*, *I* = 140 mg/L, *t*, *T* = 46 °C; (**c**)—*C*, *I* = 140 mg/L, *t* = 4 min, *T*; (**d**)—*C* = 2.2, *I*, *t*, *T* = 46 *°*C; (**e**)—*C* = 2.2, *I*, *t* = 4 min, *T*; (**f**)—*C* = 2.2, *I* = 140 mg/L, *t*, *T*.

**Figure 4 nanomaterials-12-02413-f004:**
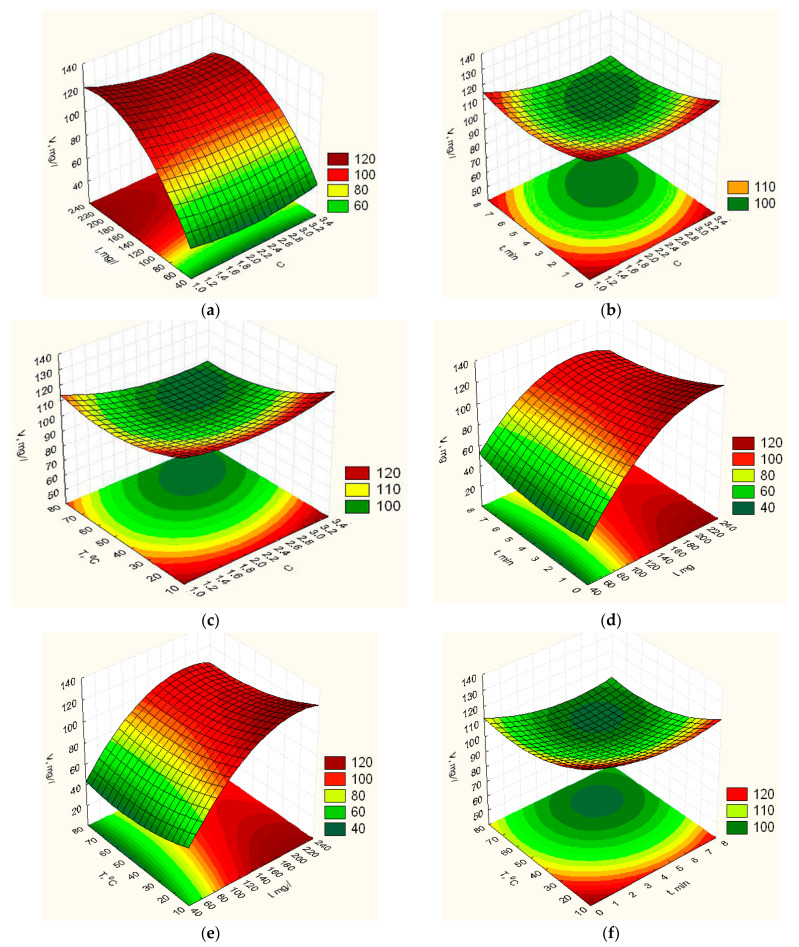
Response surfaces of dependence of the isolation of ZnO NPs from film on the composition and technological parameters of its manufacture: (**a**)—*C*, *I*, *t* = 4 min, *T* = 46 °C; (**b**)—*C*, *I*
*=* 140 mg/L, *t*, *T* = 46 °C; (**c**)—*C*, *I* = 140 mg/L, *t* = 4 min, *T*; (**d**)—*C* = 2.2, *I*, *t*, *T* = 46 *°*C; (**e**)—*C* = 2.2, *I*, *t* = 4 min, *T*; (**f**)—*C* = 2.2, *I* = 140 mg/L, *t*, *T*.

**Figure 5 nanomaterials-12-02413-f005:**
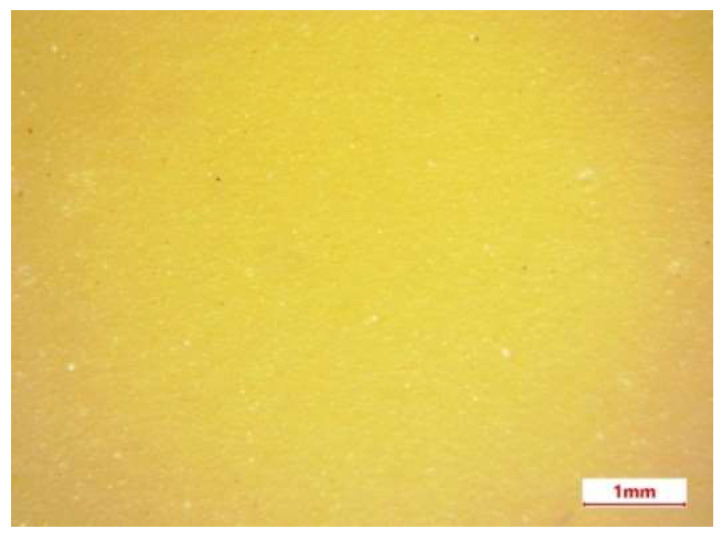
Optical micrographs of the biopolymer nanofilled films formed according to the optimal technological parameters of the process.

**Figure 6 nanomaterials-12-02413-f006:**
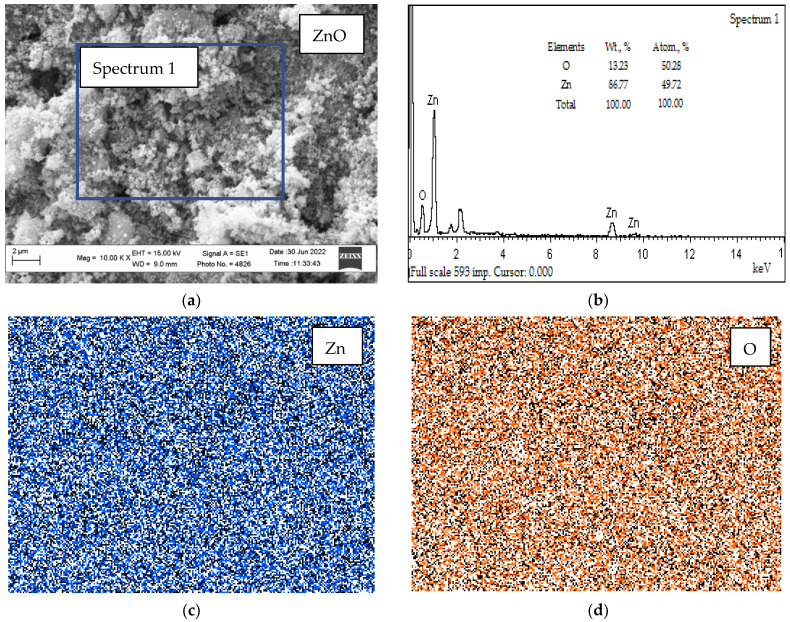
The results of the EDS analysis of ZnO NPs: (**a**)—the morphology of the ZnO NPs; (**b**)—overall spectrum ZnO NPs; (**c**)—distribution of Zn; (**d**)—distribution of O.

**Figure 7 nanomaterials-12-02413-f007:**
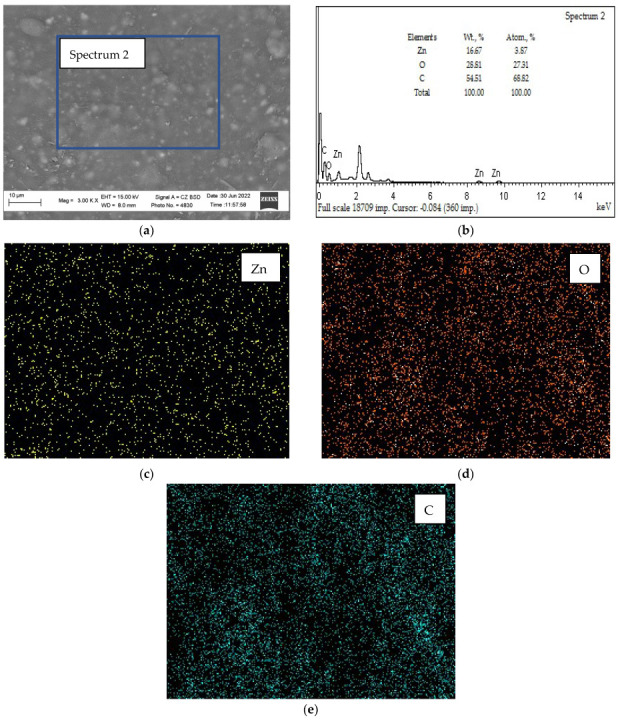
The EDS analysis results of the film surface: (**a**)—the film microstructure; (**b**)—overall spectrum; (**c**)—distribution of Zn; (**d**)—distribution of O; (**e**)—distribution of C.

**Figure 8 nanomaterials-12-02413-f008:**
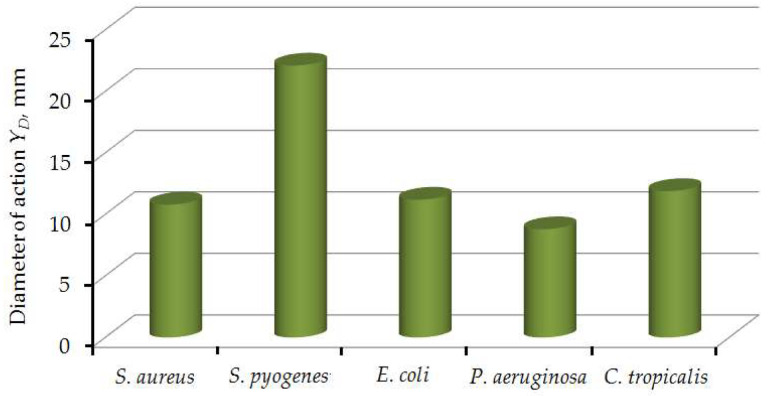
Diameter of the protective effect of biopolymer nanofilled films on the studied bacteria.

**Table 1 nanomaterials-12-02413-t001:** Coding of factors and levels of their variation during experiments.

Levels of Factors	Coded Values				Natural Values			
	*x* _1_	*x* _2_	*x* _3_	*x* _4_	Concentration Ratio (G/V), *C*	The Content of ZnO NPs, *I*, mg/L	Exposure Time in the Furnace, *t*, min	Heating Tempe- rature, *T*, °C
The main level	0	0	0	0	2.2	140	4.0	46
Interval of variation	1	1	1	1	0.7	60	2.0	18
The upper level	+1	+1	+1	+1	2.9	200	6.0	64
The lower level	−1	−1	−1	−1	1.5	80	2.0	28
Star points (+)	+1.4826	+1.4826	+1.4826	+1.4826	3.2	230	7.0	73
Star points (–)	−1.4826	−1.4826	−1.4826	−1.4826	1.2	50	1.0	19

**Table 2 nanomaterials-12-02413-t002:** Optimal values of technological parameters of the film formation process.

Optimization Parameters		Technological Parameters			
Name	Optimal Value	Concentration Ratio (G/V), *C*	The Content of ZnO NPs, *I*, mg/L	Exposure Time in the Furnace, *t*, min	Heating Temperature, *T*, °C
		Minimum Values			
		1.2	50.0	1.0	19.0
		Optimal Values			
Elasticity *Y_E_*, %	87.6	1.6	172.5	5.0	40.6
Vapour permeability *Y_P_*, g/(cm^2^ h)	407.4	2.2	117.7	7.0	59.3
Degradation time *Y_td_*, h	44.4	1.9	227.9	3.2	32.2
Isolation of ZnO NPs *Y_V_*, mg/L	114.4	3.2	178.3	7.1	72.8
Swelling *Y_H_*, %	327.4	1.4	185.7	6.1	58.2
–		**Maximum Values**			
		3.2	230.0	7.0	73.0

**Table 3 nanomaterials-12-02413-t003:** The values of optimization parameters are calculated: *Y_E_*, *Y_P_*, *Y_td_*, *Y_H_* according to the optimal values of technological parameters of film formation at which the maximum isolation of ZnO NPs *Y_V_*.

Optimization Parameters		Technological Parameters				Optimization Parameters Deviation, %
Name	Optimal Value	Concentra- tion Ratio (G/V), *C*	The Content of ZnO NPs, *I*, mg/L	Exposure Time in the Furnace, *t*, min	Heating Temperature, *T*, °C	
		Optimal Values				
		3.2	178.3	7.1	72.8	
		The Values of the Optimization Parameters are Calculated				
Elasticity *Y_E_*, %	87.6	105.8				17.2
Vapor permeability *Y_P_*, g/(cm^2^ h)	407.4	467.3				12.8
Degradation time *Y_td_*, h	44.4	56.4				21.3
Isolation of ZnO NPs *Y_V_*, mg/L	114.4	114.4				0
Swelling *Y_H_*, %	327.4	417.1				21.5

## Data Availability

Not applicable.

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
