# Peer review of "Optimization of Technological Parameters of the Process of Forming Therapeutic Biopolymer Nanofilled Films"

_nanomaterials, 2022, doi:10.3390/nano12142413_

Round 1

Reviewer 1 Report

The manuscript lack novelty and significance.

Author Response

According to the recommendations of reviewers, the article was corrected.

Reviewer 2 Report

Authors present results of an extended and quite neat study on preparation of biopolymer films for healing purposes. The topic is interesting and the study was well designed and perform. The problem is in the way of results presentation. The paper consists, almost entirely, of colorful plots of response surfaces (there is 6 figures with 6x6 plots). In my view this very discouraging for potential readers as nobody wants to go through huge amnount of (quite uninteresting) graphs and through tables full of statistical data. The statistics is powerful tool but only tool not the merit of the paper.

In my view the paper needs deep revision aiming to remove statistical data and graphs and finding a proper way of data presentation. The paper can be made substantially shorter withou loosing its strength.

Author Response

The obtained second-order dependencies are expressed in natural quantities. This makes them difficult for readers to perceive, analyze and evaluate the influence of individual factors on optimization parameters (elasticity YE, vapor permeability YP, degradation time Ytd, isolation of zinc oxide nanoparticles YV, swelling YN, and diameter of action YD). Therefore, in our opinion, it is advisable to leave the graphs in the text of the article to understand the influence of technological factors on the properties of the film and better perception of information.

Appendix A (Table A.1) presents the results of the experiments used to build the mathematical models of the second order.

Reviewer 3 Report

The article "Optimization of Technological Parameters of the Process of Forming Therapeutic Biopolymer Nanofilled Films" describes statistically a series of values obtained in the characterization of some (not presented) sample films that can be used in wound healing.

The English language needs some polishing for style and typos (e.g. “the component composition” just at the beginning of the abstract). Please be careful with terminology (e.g. “nanopolymer composites” is not the same thing as “polymer nanocomposite”). The bacterial stains across the manuscript must be written with italics.

Please try to condense the abstract. Is a bit too long. In the same direction, after first use, for example zinc oxide (ZnO), use the common abbreviations. In case of multiple citations, instead of [10,11,12,13,14,15] use [10-15].

Why authors chosen this system? Motivation must be clear. Authors should better explain the motivation behind chosen this system. The whole introduction must be reworked based on actual substances used: ZnO nanoparticles, gelatin, PVA, lactic acid etc. The accent must be put on antibacterial films that can be used in wound healing. In introduction a stronger recent literature survey is necessary, especially on previous literature reports on the biopolymer films used in wound healing and antibacterial ZnO nanoparticles. The author need to update the introduction by citing following doi: 10.3390/pharmaceutics13071020; doi: 10.3390/ma13235452; doi: 10.3390/pharmaceutics13070957 and doi: 10.1016/j.ijpharm.2013.11.035

Section 2.1 must be updated with actual work flow: origin of the substances, mixing ratio, clear steps (I don’t suppose the authors have dropped all substances at once in a beaker and start mixing them), mixing time, temperature, microwave power etc. All experimental conditions must be provided so other authors can replicate the experiment.

Section 2.2.4 is unclear to this reviewer. The authors studied the release of ZnO or Zn2+ ions? How did the authors determined on a scale of color intensity Zn2+ concentration when it has no color?

In Table 2 concentration ratio (g/va) means what? Who is va; and g is gram or what? If g and va are two of the components, then only their ratio counts? Why adding the other components then?

At least a supplementary file should be made with actual values obtained for each parameter: the vapor permeability, swelling values, antibacterial activity, elasticity etc. The authors have statistically studied a set of unavailable values, which are totally unknown to the readers.

First part of Discussion fits better to Introduction.

How is the studied system a better one? Point out the impact and usefulness of each component. Conclusion section must be reworked to underline the novelty and advantages of this research, with actual numbers.

Author Response

The article "Optimization of Technological Parameters of the Process of Forming Therapeutic Biopolymer Nanofilled Films" describes statistically a series of values obtained in the characterization of some (not presented) sample films that can be used in wound healing.

Answer

In response to comments in the article added:

We formed biopolymer nanofilled films (Fig. 8) according to the obtained optimal values of technological options.

Figure 8. Biopolymer nanofilled films formed according to the optimal technological parameters of the process.

The English language needs some polishing for style and typos (e.g. “the component composition” just at the beginning of the abstract). Please be careful with terminology (e.g. “nanopolymer composites” is not the same thing as “polymer nanocomposite”). The bacterial stains across the manuscript must be written with italics.

Answer

We agree, we have been corrected terminology in the text to the same type. Bacterial stains in the manuscript are corrected in italics. A mechanical error was made.

Please try to condense the abstract. Is a bit too long. In the same direction, after first use, for example zinc oxide (ZnO), use the common abbreviations. In case of multiple citations, instead of [10,11,12,13,14,15] use [10-15].

Answer

We agree. Amendments have been made to the text of the article (zinc oxide (ZnO) nanoparticles (NPs).

Why authors chosen this system? Motivation must be clear. Authors should better explain the motivation behind chosen this system. The whole introduction must be reworked based on actual substances used: ZnO nanoparticles, gelatin, PVA, lactic acid etc. The accent must be put on antibacterial films that can be used in wound healing. In introduction a stronger recent literature survey is necessary, especially on previous literature reports on the biopolymer films used in wound healing and antibacterial ZnO nanoparticles. The author need to update the introduction by citing following doi: 10.3390/pharmaceutics13071020; doi: 10.3390/ma13235452; doi: 10.3390/pharmaceutics13070957 and doi: 10.1016/j.ijpharm.2013.11.035

Answer

We chose a system of biopolymer film based on gelatin, polyvinyl alcohol, water, lactic acid, glycerin with ZnO NPs for research. This system is formed from available and widely used components, high-tech in production. It provides easy release of active substance ZnO NPs when used in medical practice, is biodegradable and does not lead to environmental pollution.

The review has been corrected to take into account the comments.

Section 2.1 must be updated with actual work flow: origin of the substances, mixing ratio, clear steps (I don’t suppose the authors have dropped all substances at once in a beaker and start mixing them), mixing time, temperature, microwave power etc. All experimental conditions must be provided so other authors can replicate the experiment.

Answer

We choose the following components for biopolymer nanofilled films based on literature review and the experience of our own previous studies: gelatin, polyvinyl alcohol, distilled water, lactic acid, glycerin and zinc nanooxide.

To form prototypes of biopolymer nanofilled films, we used chemically pure components produced by gelatin - Weishardt International, Slovakia; polyvinyl alcohol - Sure Chemical Co., LTD. Shijiazhuang; lactic acid solution (80%) - Cosco, China; Glycerin - VG, Gerimpeks, Germany; zinc oxide nanopowder (20-30 nm) –Hingwunewmaterial, China. We used Aquadistillator DE-4-2 to obtain distilled water.

For studying the therapeutic effect of wound-healing polymer material and its physical and mechanical properties depending on the component composition and technological parameters of the process of formation of this material used the following components: gelatin, polyvinyl alcohol, distilled water, lactic acid, glycerin, and zinc oxide nanoparticles with a particle size of 30 nm.

The proposed wound-healing polymer material was obtained according to the developed technological process (Fig. 1). , which contains the following basic operations:

− weighing of components;

− mixing of components;

− microwave heating of the mixture to a given temperature;

− mixing of components;

− exposure at a given temperature;

− pouring the mixture into a mold to obtain a film;

− film quality control.

Figure 1. Block diagram of the technological process of formation of  biopolymer nanofilled films

We weighed the components used AXIS AD200 balance with an accuracy of 0.001 g. We took the original composition of the film according to the patent UA110594U. We varied only the ratio of the amounts of gelatin to polyvinyl alcohol and the mass fraction of zinc nanooxide, and the mass fractions of the other components remained unchanged. We also varied temperature and exposure time during the experiments according to the plan. We heated the mixture in a microwave oven SAMSUNG GE88SUB / UA (maximum power of ultra-high frequency radiation 800 W).

At the first stage of preparation of the mixture, gelatin, polyvinyl alcohol and water were added to the fluoroplastic beaker and stirred with a glass rod. The resulting mixture was heated to a given temperature at a high-frequency radiation power of 300 W, stirring it periodically every 30 seconds.

At the second stage, stirring the mixture, alternately a solution of lactic acid (80 %) and glycerin was added and again heated to a given temperature at a power of ultra-high frequency radiation of 300 W.

At the third stage, stirring the mixture, nanopowder of zinc oxide was added and subsequently kept in a microwave oven at a power of 150 W for an appropriate period of time according to the experimental plan.

Next, the mixture was poured into a mold with a chrome coating and kept to complete the polymerization process for 24 hours at room temperature. The formed film was removed from the mold and its quality was visually inspected using a seven-fold Attache MG1004A magnifier.

Section 2.2.4 is unclear to this reviewer. The authors studied the release of ZnO or Zn2+ ions? How did the authors determined on a scale of color intensity Zn2+ concentration when it has no color?

The study of the isolation of ZnO NPs from a polymer film was carried out using a calorimetric test system for zinc Aquaquant® company Merck KGaA (Germany) with a sensitivity of 0.1–5 mg/l, under the conditions of obtaining the film by the method of formation of Zn2 + complexes with film components (gelatin, olivinyl chloride, lactic acid. acid - known as complexones).

Samples of the films were placed in chemical beakers of distilled water, and samples of the solution were taken after 5, 15, 30, 60 min, and 24 hours, respectively. Detection of Zn2 + was performed on a colored blue-green complex formed by the reaction of the selected solution with thiocyanate ions in the presence of diamond green. The calibration graph was based on samples with well-known values of ZnO content in the film.

The concentration of zinc ions was determined on a scale of colour intensity, which corresponded to specific concentrations. Based on the obtained results of zinc concentration, zinc oxide evolution was determined taking into account the known ratio ZnO / Zn = 81.408 / 65.38.

We took the number of isolated selected ZnO NPs for 24 hours in the calculations.

In Table 2 concentration ratio (g/va) means what? Who is va; and g is gram or what? If g and va are two of the components, then only their ratio counts? Why adding the other components then?

Answer

The main components of the film are gelatin and polyvinyl alcohol, so to reduce the number of factors during the planning of the experiment used the ratio of components C / V (gelatin / polyvinyl alcohol). The article replaces g/va with C/V

The original composition of the film was adopted on the basis of patent UA110594U [Popadyuk]. We varied only the ratio of the amounts of gelatin to polyvinyl alcohol and the mass fraction of ZnO NPs, and the mass fractions of the other components remained unchanged. Without the other components, the formation of a film with high physico-mechanical and therapeutic properties (lactic acid - a crosslinking component, glycerin - a emollient component, ZnO NPs - the active substance) is not provided.

At least a supplementary file should be made with actual values obtained for each parameter: the vapor permeability, swelling values, antibacterial activity, elasticity etc. The authors have statistically studied a set of unavailable values, which are totally unknown to the readers.

Answer

In response to comments we added.

Table А.1. Plan-matrix of experimental results

Experiment number

Levels of factors

The average value of the optimization parameters

х0

х1

х2

х3

х4

Yav

Elasticity YЕ, %

Vapour permeability

YP, g/(cm2hrs)

Degradation time Ytd, hrs

Isolation of zinc oxide nanoparticles YV, mg

Swelling YН, %

1

+1

–1

–1

–1

–1

Yav01

75.9

461.0

36.9

77.0

327.2

13.0

2

+1

–1

–1

–1

+1

Yav02

81.8

456.8

46.8

75.2

321.6

12.8

3

+1

–1

–1

+1

–1

Yav03

77.9

427.4

42.8

80.6

310.1

13.4

4

+1

–1

–1

+1

+1

Yav04

79.0

380.2

47.6

77.0

288.3

13.0

5

+1

–1

+1

–1

–1

Yav05

73.4

466.2

43.7

127.4

294.7

18.6

6

+1

–1

+1

–1

+1

Yav06

69.1

414.0

48.2

121.1

291.7

17.9

7

+1

–1

+1

+1

–1

Yav07

71.2

427.9

47.5

118.4

281

17.6

8

+1

–1

+1

+1

+1

Yav08

67.6

363.3

49.7

112.1

268.3

16.9

9

+1

+1

–1

–1

–1

Yav09

83.9

489.3

45.2

79.7

365.6

13.3

10

+1

+1

–1

–1

+1

Yav10

88.1

480.3

53.1

72.5

332

12.5

11

+1

+1

–1

+1

–1

Yav11

80.4

416.9

47.0

79.7

338.4

13.3

12

+1

+1

–1

+1

+1

Yav12

85.2

395.5

52.6

71.6

295

12.4

13

+1

+1

+1

–1

–1

Yav13

76.3

459.9

47.3

124.7

347.9

18.3

14

+1

+1

+1

–1

+1

Yav14

75.6

433.6

52.5

113.0

323.4

17.0

15

+1

+1

+1

+1

–1

Yav15

74.0

413.4

49.8

112.1

330.3

16.9

16

+1

+1

+1

+1

+1

Yav16

74.1

372.8

52.8

101.3

296

15.7

17

+1

–1.4826

0

0

0

Yav17

72.7

421.0

45.4

106.7

283.5

16.3

18

+1

+1.4826

0

0

0

Yav18

79.2

437.5

51.5

102.2

326.3

15.8

19

+1

0

–1.4826

0

0

Yav19

88.6

365.1

41.2

49.1

310.7

9.9

20

+1

0

+1.4826

0

0

Yav20

74.3

350.4

44.3

108.5

302.4

16.5

21

+1

0

0

–1.4826

0

Yav21

76.9

452.4

45.2

107.6

323.4

16.4

22

+1

0

0

+1.4826

0

Yav22

73.1

355.6

47.7

100.3

285.4

15.6

23

+1

0

0

0

–1.4826

Yav23

69.7

436

43.6

113.9

320.2

17.1

24

+1

0

0

0

+1.4826

Yav24

70.0

386.9

51.2

95.9

285.5

15.1

25

+1

0

0

0

0

Yav25

73.3

394.6

44.4

99.5

288.4

15.5

26

+1

0

0

0

0

Yav26

73.3

394.6

44.4

99.5

288.5

15.5

First part of Discussion fits better to Introduction.

Answer

We agree with the comment and a part of the text from point 4. Discussion we moved to point 1. Introduction.

How is the studied system a better one? Point out the impact and usefulness of each component. Conclusion section must be reworked to underline the novelty and advantages of this research, with actual numbers.

Answer

For research, the selected system of biopolymer film is formed from available and widely used components, high-tech in manufacturing, provides easy release of the active substance ZnO NPs, convenient for use in medical practice, is biodegradable and does not pollute the environment. This article optimizes the technology of obtaining biopolymer nanofilled films, which allowed to improve its physical, mechanical and therapeutic properties compared to the results of previous studies doi: 10.18484 / 2305-0047.2019.1.16.

Regarding the effect and usefulness of each component of biopolymer nanofilled films: gelatin – a water-soluble crosslinking and polymerizing agent of polyvinyl alcohol; polyvinyl alcohol – water-soluble polymer of linear structure; distilled water – solvent; lactic acid – a crosslinking component; glycerin – emollient component; zinc nanooxide is the active substance.

According to the results of research on the influence of component composition and technological parameters of the process of forming biopolymer nanofilled films:

  1. For the first time multifactor mathematical models of the second order for determination of physical-mechanical and medical properties are constructed (the elasticity YЕ, vapor permeability YP, degradation time Ytd, isolation of ZnO NPs YV, swelling YН and diameter of action YD) on the concentration ratio gelatin to vinyl alcohol C, the content of ZnO NPs I, exposure time in the furnace t and heating temperature T were built;
  2. Based on the obtained mathematical models, the optimal values of the parameters of the process of formation films (component composition and technological parameters) were established to ensure optimal characteristics of the polymer film (the elasticity YЕ, vapor permeability YP, degradation time Ytd, isolation of ZnO NPs YV, swelling YН and diameter of action YD) (Table 3);
  3. It was established that the response surfaces of the dependence of the antibacterial action of wound-healing biodegradable nano-containing polymer material biopolymer nanofilled films, which was evaluated by the diameter of action, on the composition and technological parameters of its formation are similar to the isolation of ZnO NPs from this material films (Table 3);

Round 2

Reviewer 1 Report

The manuscript by Bembenek et al "Optimization of Technological Parameters of the Process of Forming Therapeutic Biopolymer Nanofilled Films" requires revision before its publication.

 Comments

1. All abbreviations should be cross-varified in the text.

2. Introduction section is weak. Too much repeating information with large numbers of citations. Authors should more precisely revise this section into 3 to 4 paragraphs only i.e. i) Instead of "the article [19]", and "authors [50]"  use appropriate authors' names of citations to better scientific presentation; ii) Lines 58-64, please also add information on the therapeutic application polymers-based biotechnological applications such as biocatalysts i.e doi:10.3390/polym14071409; iii) please add information on the bio-based polymers with few examples such polyhydroxyalkanoates (degradable biopolymers) and its comp[osite to NPs for therapeutic applications such as antimicrobial agents and tissue engineering i.e. doi: 10.1016/j.biortech.2021.124737; iv) mechanism of antimicrobials in the present study; v) please avoid too many citations in sentences, only use two-three citations in such all cases throughout the manuscript. 

3. Materials and methods: many experimental details have been known. Such details may be minimized by appropriate citations.

4. Results, and Discussion: the antimicrobial data can be provided and discussed in brief. How it can be effective for a broad range of organisms such as bacteria and fungi (section 2.2,6). 

5. Provide the instrumental data to confirm the presence of NPs such as EDS (elemental mapping data).

Author Response

Dear Reviewer,

the replies are supplemented in the pdf file.

Best regards,

Authors

Reviewer 2 Report

I insist on my comments to the original manuscript.

Author Response

Dear Reviewer,
the answers are supplemented in the pdf file.
Best regards,
Authors

Reviewer 3 Report

The authors have responded to my comments and have addressed all my concerns, greatly improving the manuscript, therefore, I suggest publishing the paper in the current form.

Author Response

The team of authors express their gratitude to the reviewer 3 for valuable recommendations that have been taken into account to improve significantly the quality of this paper.

Round 3

Reviewer 1 Report

The manuscript has been improved. It can be accepted.